

# Periwinkle climbing response to water- and airbone predator chemical cues may depend on home-marsh geography

John M. Carroll[1,2,*], Morgan B. Church[2,*] and Christopher M. Finelli[2]

[1] Department of Biology, Georgia Southern University, Statesboro, GA, USA
[2] Department of Biology and Marine Biology, University of North Carolina Wilmington, Wilmington, NC, USA
* These authors contributed equally to this work.

## ABSTRACT

The salt marsh periwinkle, *Littorina irrorata*, exhibits a spatial refuge from predation by climbing the stems of *Spartina alterniflora* in order to avoid benthic predators. Salt marsh periwinkles have a broad geographic distribution, and for many species, responses to predators also varies with biogeography. This study sought to determine if the geographical location of the home marsh influenced the response of periwinkles (climbing height) to blue crab predator cues both via air and water. Snails from Louisiana (LA) climbed higher in general than those from North Carolina (NC), regardless of chemical cue. However, LA snails climbed 11 cm higher in the presence of waterborne predators than control snails with no cue, while NC snails only climbed five cm higher in the same comparisons. Airborne chemical cue tended to have snails climbing at intermediate heights. These responses were significantly enhanced when both populations of snails were housed together. Periwinkle response to predator cues was stronger in LA than NC, and so it is possible that the behavioral response of these snails to predators varies with biogeography of the home marsh. Also interestingly, the results of this study also suggest that cue delivery is probably occurring via mechanisms other than water, and potentially via airborne cues. Therefore, salt marsh periwinkles likely respond to numerous cues that initiate behavioral responses, including airborne cues, and these responses may vary by home-marsh geography.

# INTRODUCTION

Predation is one of the most important interactions affecting marine populations (*Connell, 1975*; *Behrens Yamada, Navarrete & Needham, 1998*). Predators can directly affect the distribution, abundance, size structure and genetic make-up of prey populations (*Menge, 1983*; *Yoshida et al., 2003*). As a result of intense predation pressure, prey have evolved various means to reduce predation risk that vary on ecological and evolutionary timescales (*Vermeij, 1982*; *Trussell & Smith, 2000*). For example, natural selection is thought to drive changes in prey morphology over evolutionary timescales, with prey

Corresponding author
Christopher M. Finelli,
finellic@uncw.edu

growing thicker, more ornate exoskeletons in response to high or increasing predation pressure (*Vermeij, 1982*, *1983*, *1987*). However, prey can respond to predators at ecological (within lifetime) timescales (*Lima & Dill, 1990*). In particular, predators have increasingly been demonstrated to rapidly induce prey defenses, which act to reduce prey vulnerability (*Trussell & Smith, 2000*). These inducible defenses occur across diverse taxa and include fast growth, chemical defenses, skeleton thickening, changes in behavior, and using refugia (*Harvell, 1990*; *Berenbaum & Zangerl, 1999*).

A number of gastropods have demonstrated inducible defenses as a result of predation pressure in experimental settings (*Behrens Yamada, Navarrete & Needham, 1998*; *Brandwood, 1985*; *Duncan & Szelistowski, 1998*; *Large & Smee, 2010*, *2013*). A common defense is changing behavior, including predator avoidance and/or fleeing (*Legault & Himmelman, 1993*). However, these defenses typically vary across broad georgraphic scales. Both predator diversity and predation pressure generally vary inversely with latitude, so prey organisms have responded by increasing defenses with decreasing latitude (*Laurila, Lindgren & Laugen, 2008*; *Freestone et al., 2011*), which includes latitudinal differences in behavioral responses (*Aschaffenburg, 2008*; *Donahue et al., 2009*; *Duval, Calzetta & Rittschof, 1994*; *Large & Smee, 2013*). Induced defenses are affected at broad biogeographic scales by differences in environmental conditions and stimuli (*Trussell & Smith, 2000*). Further, there are costs associated with induced defenses (*Trussell & Nicklin, 2002*), so geographic patterns in prey response likely reflect the greater predictability of predation risk at certain locations (*Trussell & Smith, 2000*).

For intertidal snails, predator avoidance includes using spatial refugia to avoid capture which has been demonstrated in both rocky-interidal (*Menge & Lubchenco, 1981*) and salt marsh habitats (*Warren, 1985*). The salt marsh periwinkle, *Littorina irrorata* Say, is an important resident of tidal marshes along the US Atlantic and Gulf coasts, which exhibit spatially variable distribution dependent upon the tidal stage (*Hovel, Batholomew & Lipcius, 2001*). Historically, the distribution of periwinkles in the salt marsh was initially considered to be the result of environmental variables (*Bingham, 1972*). However, considerable evidence suggests this vertical distribution is to avoid predators when the tide returns, such as the blue crab, *Callinectes sapidus* Rathbun, and the crown conch, *Melongena corona* Gmelin (*Hamilton, 1976*; *Warren, 1985*), and perwinkles tend to migrate higher and/or faster in the presence of predators (*Dix & Hamilton, 1993*; *Duval, Calzetta & Rittschof, 1994*; *Kimbro, 2012*; *Wollerman, Duva & Ferrier, 2003*), although these activities are constrained by environmental stressors, such as temperature (*Iacarella & Helmuth, 2012*). Thus, periwinkles exhibit a spatial refuge from predation by climbing the stems of salt marsh cordgrass, *Spartina alterniflora* Loisel, in order to avoid benthic predators (*Dix & Hamilton, 1993*; *Vaughn & Fisher, 1988*).

*Littorina irrorata* has an extensive geographic range (*Bingham, 1972*) and climbing behavior has been noted at the local scale in Virginia (*Stanhope, Banta & Temkin, 1982*), North Carolina (NC) (*Duval, Calzetta & Rittschof, 1994*; *Lewis & Eby, 2002*), Georgia (*Silliman & Bertness, 2002*), Florida (*Hamilton, 1976*; *Warren, 1985*), Alabama (*Henry, McBride & Williams, 1993*), Louisiana (LA) (C. Finelli, 2006, personal observation), and Texas (*Vaughn & Fisher, 1988*). Since predator diversity and predation

pressure vary with latitude (*Bertness, Garrity & Levings, 1981*; *Freestone et al., 2011*), and salt marsh periwinkles inhabit this broad geographic range, they are useful model organisms to explore biogeographic variation in behavioral responses. Regional comparisons in climbing height and response to predators in marsh periwinkles have not been made previously, although a number of other similar species have exhibited differential responses to predators across geographic ranges (*Fawcett, 1984*; *Large & Smee, 2013*). Therefore, periwinkles might exhibit similar differences in induced avoidance responses according to home-marsh geography.

The mechanism thought to be driving climbing behavior is response to waterborne chemical cues from either predators or injured conspecifics (*Duval, Calzetta & Rittschof, 1994*; *Jacobsen & Stabell, 1999*), although periwinkles often migrate in advance of the tide. For other intertidal snails, such as *Cerithidea decollata*, it has been suggested that there is an internal clock that might drive migrations (*Lazzeri et al., 2014*), however *L. irrorata* has been demonstrated to rapidly respond to out of phase tidal cycles in the lab (*Hovel, Batholomew & Lipcius, 2001*). It is possible that some cues might become aerosolized prior to the arrival of the incoming tide, forewarning snails and cueing them to start climbing (*Lazzeri, 2017*). A number of terrestrial gastropods respond to airborne cues for homing (*Chelazzi, Le Voci & Parpagnoli, 1988*), feeding (*Davis, 2004*), mating (*Ng et al., 2013*), and avoiding predators (*Bursztyka et al., 2013*). Interestingly, it has been suggested that at least two species of intertidal snails may respond to airborne cues from either food (*Fratini, Vannini & Cannici, 2008*) or the environment (*Lazzeri, 2017*). Given the responses to other airborne chemical cues, it is possible that intertidal marine gastropods would also react to airborne cues from predators, particularly snails such as *L. irrorata*, which spend much of their time emersed. Yet responses to potential airborne chemical cues from predators have not been investigated in Littorinids.

Relatively little is known about how species might respond to different chemical cues across geographical locations. Due to its geographic range and behavior, the marsh periwinkle is a useful model organism to explore whether geographic location and the presence of airborne cues affects anti-predator behavioral responses. Further, periwinkles could also be a useful model organism to see whether behavioral responses change in the presence of individuals from different populations, which may come into contact due to human activities. Field observations in LA demonstrated that periwinkles responded to crabs by climbing up *S. alterniflora*, however, similar field observations in NC suggested a lesser response. Thus, we designed a controlled lab experiment to investigate the difference in behavioral response (climbing) of two periwinkle populations to cues from a common predator, the blue crab *C. sapidus*. Specifically, we tested whether the presence of both waterborne and airborne blue crab cues would cause snails to migrate up *Spartina* mimics, and whether the two populations would climb to different heights. Since behavioral responses to predation are likely to vary at different geographic locations and predation pressure often increases with decreasing latitude, we hypothesized the LA population of periwinkles would show a greater response to the predator than the NC population by migrating higher on the mimics. Additionally, since intertidal salt marsh periwinkles spend the majority of their time out of the water,

we hypothesized that airborne cues would elicit a behavioral response, although given the marine origin of this species, we expected the airborne response to be intermediate.

## METHODS AND MATERIALS

Louisiana snails were collected from *S. alterniflora* marsh adjacent to the Louisiana Universities Marine Consortium (29°15′20.65″N, 90°39′42.93″W) and transported to NC. NC snails were collected from the salt marsh adjacent to the Center for Marine Sciences (34°08′26.26″N, 077°51′47.81″W). These locations were chosen because they are within the range of *L. irrorata* and they share an abundant common predator, blue crabs. In order to acclimate snails to laboratory conditions, individuals were held in the lab for 2 days prior to the start of the experiment since the species has been shown to rapidly (~1 day) respond to changes in tidal conditions in the lab (*Hovel, Batholomew & Lipcius, 2001*). Snails were provided with *Spartina* as a food resource during the holding period. Blue crabs were purchased from a local supplier. All animals were kept in flowing filtered seawater holding tanks. Crabs were fed crushed snails ad libitum for 48 h prior to use in experiments.

Lab assays were conducted to investigate regional differences in climbing and to test for chemical cue responses. The experimental unit was a box-in-box mesocosm set-up (Fig. 1A). Briefly, we placed a small plastic tub (27 × 41 × 18.5 cm), used to house the periwinkles during the experiment, within a larger plastic tub (39 × 54 × 16 cm). *Spartina*-mimics were used to simulate marsh grass in lab assays (*Hovel, Batholomew & Lipcius, 2001*). Eight 75 cm tall × 1.5 cm wide polyvinyl chloride (PVC) pipes were used in each replicate. The PVC mimics were preferable to natural grass because they are inert (*Sueiro, Bortolus & Schwindt, 2012*), can be easily washed between trials, and are not likely to give off chemical cues like wooden dowel rods or *Spartina* stems. In crab water cue treatments, the small, inner plastic tubs were drilled with small holes to allow water to mix between the inner and outer tubs when filled (Fig. 1A). For airborne cues and no cue treatments, the inner boxes were not drilled in order to isolate the water in the small tub. However, the airborne cue treatments held a crab in the outer box, whereas no cue treatments did not receive a crab. Plastic mesh was used to cover the space between the small and large tubs to prevent movement of animals between the tubs (Fig. 1B). We used six box-in-box set-ups per trial for the experiments. Air stones, modified to reduce splashing, were placed in the outer tub for all treatments. Each experimental unit was surrounded on four sides by a 45 × 64 × 90 cm open top cage constructed of PVC pipe and a thick black plastic curtain to isolate the replicates from each other, preventing potential transfer of airborne cues between units and reducing the visual disturbance on both snails and crabs. Fluorescent work lights were provided directly above each experimental unit. Temperature was maintained at 28 °C in holding tanks and experimental units.

Three different treatments were established—a no crab control, a crab present with chemical cues mixing via water exchange between tubs, and a crab present with no water exchange. For the no crab control, ~26 L of clean, filtered, and sterilized seawater was placed into the two tubs, for a water depth of 12.5 cm. For the crab present
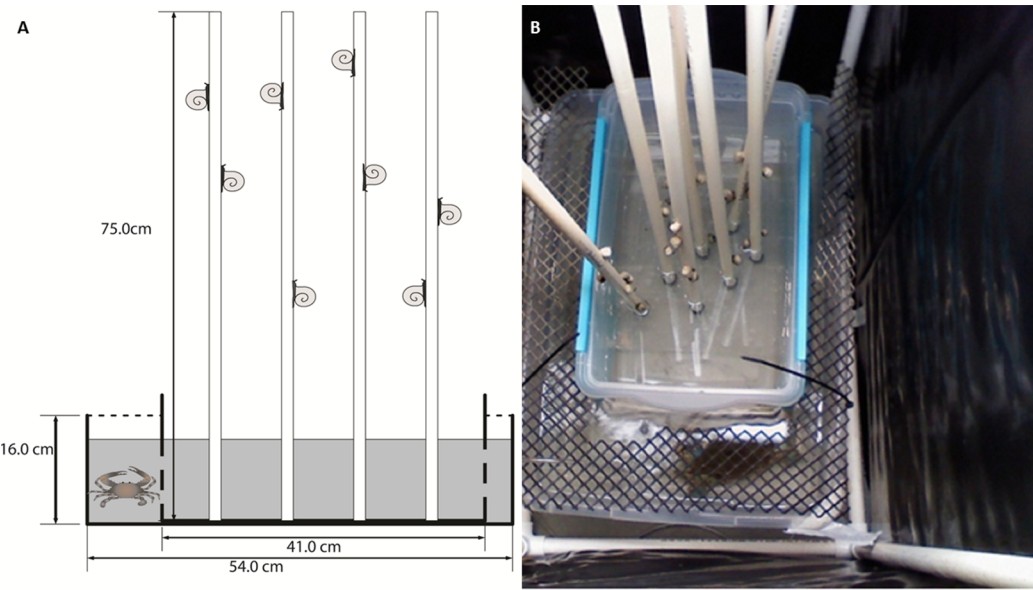

**Figure 1 Conceptual diagram and photograph of experimental box-within-a-box design.** Conceptual diagram (A) of the box-within-a-box experimental design. Snails and *Spartina* mimics were places in the inner plastic box, and when a crab was present, it was placed in the outer box. Inner boxes were either perforated to allow water exchange (as shown in A) or kept solid to prevent water exchange. Photo (B) of the experimental set-up showing the mesh screen to prevent the crab from entering snail tubs and the PVC frames and black plastic curtains that surround each experimental unit to minimize mixing of cues. Symbols courtesy of the Integration and Application Network, University of Maryland Center for Environmental Science (http://ian.umces.edu/symbols/). Photo by CM Finelli.

treatments, crabs were placed in the large outer tub and allowed to move freely throughout the space between the tubs. For the waterborne cue, the small, inner tubs drilled to allow water exchange were used, circulating the water between tubs and allowing snails to contact water exposed to the blue crab. For the non-waterborne cue, tubs that were not drilled were used, restricting both the crab and crab-cue water to the outer tub, and thus the snails could not directly sample water with crab cues. These tubs were covered with a mesh top which allowed airborne cues to escape. Our mesocosms were undisturbed during the experiment to minimize stimulation of test subjects. We did not notice surface splashing by crabs, they were either stationary or remained submerged for the duration of the trials. Thus, any response in these treatments is indicative of an airborne cue.

Two sets of experiments were conducted. The first set used either LA or NC snails alone, and two trials were used for each of the NC and LA snails. Sets of 30 snails were placed directly on the PVC approximately five cm above the water line (~17 cm above the bottom of the tubs) in each mesocosm set-up ($n = 180$ total snails per trial) and exposed to the different treatments for 6 h. Observations of snails demonstrated that many initially approached and entered the water at the start of the experiment. Since the airborne cue was the target of the investigation, each trial had airborne cue treatments. However, due to space limitations for the experimental set-ups, waterborne cue and no cue treatment replicates were only used in a single trial.

To eliminate any perceived differences between populations that might have been due to running separate experiments, we conducted a second common garden experiment where we combined snails from the two populations. For the second experiment, we also used two trials, although to keep density per mesocosm the same, we only used 15 snails per home marsh. This allowed us to directly examine the two populations in the same experimental conditions. At the end of each trial, the height of each individual snail was measured.

For the single population experiments, NC and LA trials were combined and analyzed. A two-factor generalized linear mixed model (GLMM) was used to determine the effect of site and cue treatment on average climbing height of snails. Site (NC or LA) and cue treatment (no cue, airborne cue, waterborne cue) were modeled as a fixed effect. Since 30 snails were placed into each tub for the experiment, snails within each tubs could not be considered independent of each other (Supplemental Information 1). To account for potential effects of snails located within the same tub, a tub identifier was included as a random effect in the model. GLMMs were also used to determine the effect of home marsh and cue type in the mixed population experiment, with treatment (no cue, airborne cue, waterborne cue) and location (NC and LA) as fixed effects, and tub modeled as a random effect. Models were fit with the function "glmer" and a gamma distribution using the "lme4" package (*Bates et al., 2015*) in R (*R Core Team, 2015*). When significant effects were found, post hoc general linear hypothesis comparisons were performed using "ghlt" function "multcomp" package in R (*Hothorn, Bretz & Westfall, 2008*).

## RESULTS

Snails actively moved up and down the PVC mimics during the experimental period. LA snails climbed significantly higher than NC snails ($p < 0.001$), although there was a significant interaction between the two fixed factors ($p < 0.001$), suggesting the response in the different snail populations varied dependent upon the cue treatment (Fig. 2). There was a trend in the LA snails to climb higher when exposed to predator cues, although there was high variability within each treatment, and these trends were not significant (Fig. 2). The minimum height for waterborne cue boxes was 16.4 cm, whereas it was 19.3 for airborne cue and 22.5 for control boxes. The maximum was similar for all (74.8, 76.5, and 76.4 cm) for waterborne, airborne, and control boxes. No individuals were below the water line at the end of the experiment. The within trial variability in mean snail height among boxes of the same treatments was 5.4, 12.6, and 2.9 cm for waterborne, airborne and control treatments, respectively.

For NC snails, there was also a trend for those exposed to waterborne crab cues to climb higher than those with no cue (Fig. 2), although like in the LA snail population, there was considerable variability among individuals. Interestingly, the minimum height for NC snails was 4.8, 0, and 0 cm for the waterborne, airborne and control treatments, respectively. The maximum height climbed in the waterborne cue was 66.3 cm, whereas the maximum height in the airborne cue was 68.5 cm. The maximum height climbed in the control boxes was lower (51.6 cm). Across all control boxes, 27% of the individuals were submerged at the end of the trial, while 36% of snails in the airborne boxes

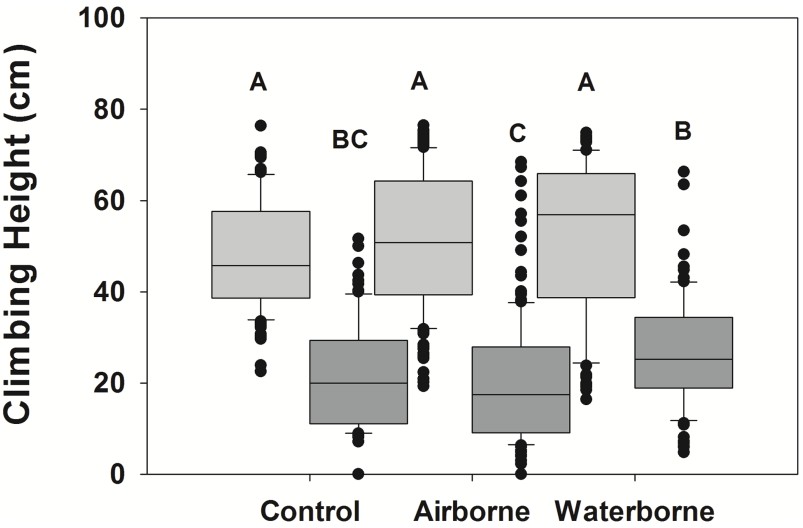

**Figure 2 Climbing height of different snail populations across different cue treatments.** Climbing height in single population assemblages for Louisiana (light gray boxes) and North Carolina snails (dark gray boxes) in the presence of no cue, an airborne cue, and a waterborne cue. The boundaries of the box represent the 25th and 75th percentiles, the solid line represents the median, the whiskers are the 10th and 90th percentiles, and the dots represent outliers. Letters above the boxes indicate significant differences in post hoc analysis.

**Table 1 Analysis of deviance table for mixed population experiments.**

|  | $\chi^2$ | D$f$ | $p$ |
|---|---|---|---|
| Treatment | 30.9221 | 2 | 1.929e-07*** |
| Site | 63.6062 | 1 | 1.520e-15*** |
| Treatment × site | 1.0741 | 2 | 0.5845 |

**Note:**
Analysis of deviance table for differences in climbing height by site (LA and NC) and cue treatment (no cue, airborne cue, waterborne cue) in the mixed population assemblage using GLMM with site and cue treatment as the fixed factors and tub as the random effect.
*** indicates $p < 0.001$.

and only 11% of snails in the waterborne boxes were submerged. As with LA snails, there was also within trial variability in mean height across boxes of the same treatments. Mean height varied by 5.3, 10.9, and 5.2 cm across boxes in the waterborne, airborne, and control treatments, respectively.

When snail populations were placed together, there were significant treatment and location effects ($p < 0.001$ for both), but no significant interaction ($p = 0.585$; Table 1). LA snails climbed significantly higher (41.4 ± 1.2 cm, mean ± SE) than NC snails (22.8 ± 1.2, $p < 0.001$) across all treatments (Fig. 3A). In addition, across both sites, snails in the airborne (38.3 ± 1.6 cm) and waterborne (36.7 ± 1.8 cm) cues climbed significantly higher than those in no cue treatments (21.8 ± 1.3; $p < 0.001$ for both). Climbing heights in the presence of airborne or waterborne cues were not different from each other ($p = 0.648$; Fig. 3B). Within the mixed treatment, LA snails in the presence of crab cues climbed between 19 and 24 cm higher on *Spartina* mimics than those not exposed to crab cues, whereas NC snails climbed between 9 and 10 cm higher when crab cues were present vs. absent.

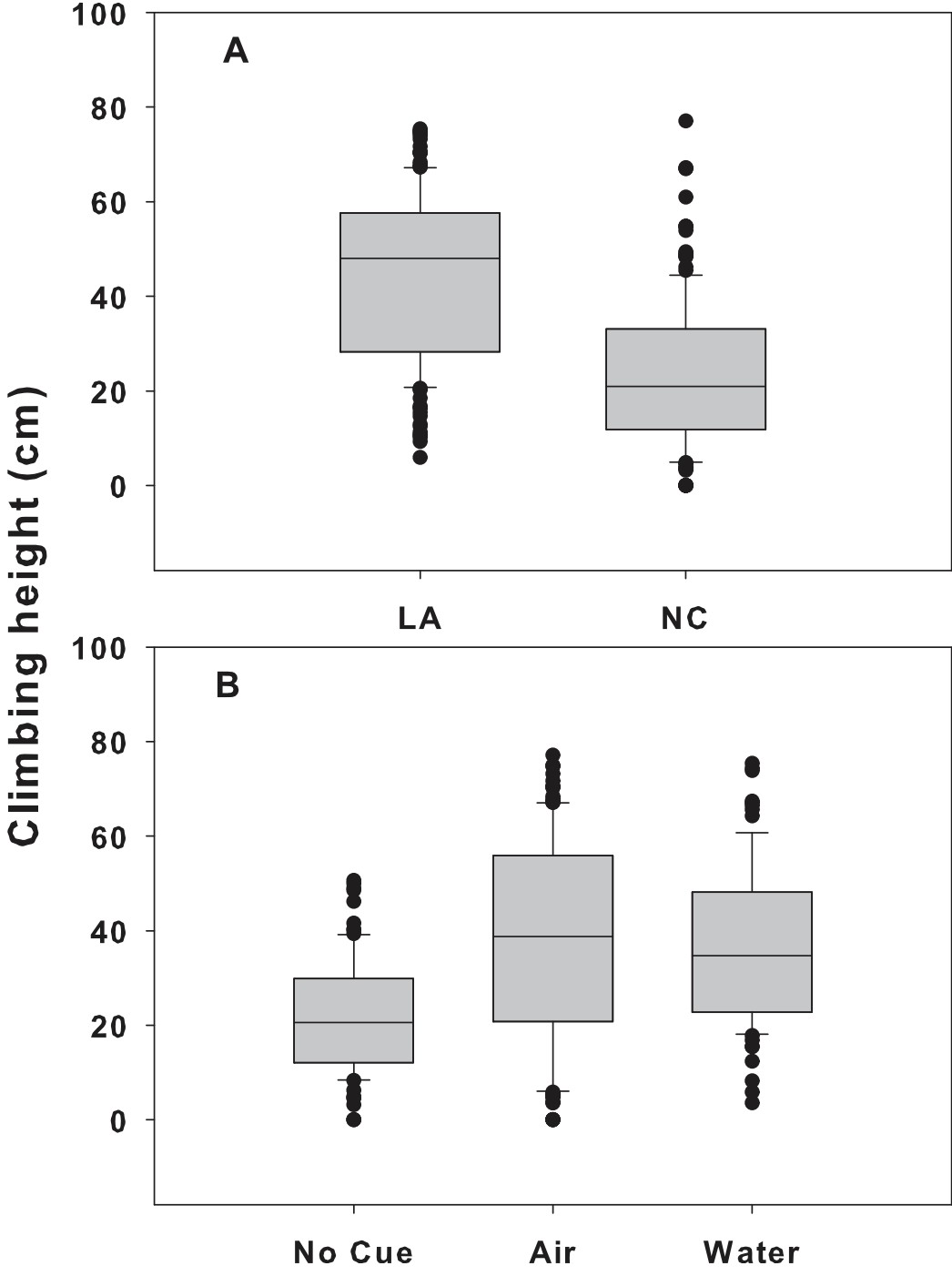

**Figure 3 Climbing height for each population and cue treatment in the mixed-population assemblages.** Differences in climbing height between the two populations (A) and across all three cue treatments (B) in the mixed-population assemblage experiment. The boundaries of the box represent the 25th and 75th percentiles, the solid line represents the median, the whiskers are the 10th and 90th percentiles, and the dots represent outliers.           

## DISCUSSION

Marsh periwinkles have an inducible defense that they exhibit over their broad geographic range, making them an ideal model species of examining geographic variation in predator avoidance behavior. In this study, salt marsh periwinkles from both populations responded to the presence of blue crabs by migrating higher up *Spartina* mimics than those in control treatments, a result consistent with earlier findings (*Warren, 1985*), although here the differences were more apparent when the snails were housed in mixed population treatments. However, it was previously unknown whether the behavioral response, in this case migration distance, might be greater in the lower latitudes. Snails from LA demonstrated a stronger response by migrating farther up the mimics when kept in both the single population and mixed assemblages than their NC counterparts. This mimicked our field observations at both LA and NC, where LA periwinkles consistently and reliably climbed up *S. alterniflora* stems in response to crab odors, including airborne cues, while those in NC were less consistent in their response. It is therefore possible that the behavioral response of *L. irrorata* to crab presence varies geographically.

Several environmental factors may have influenced marsh periwinkle vertical migration (*Bingham, 1972*; *Henry, McBride & Williams, 1993*), including tidal regime (*Kimbro, 2012*). LA snails in our experiment consistently climbed higher than NC snails, even in the absence of predator cues. The snails should have become entrained with their new conditions in the lab, since periwinkles have been demonstrated to rapidly respond to changing tidal cycles and constant water depth (*Hovel, Batholomew & Lipcius, 2001*). Further, the tidal amplitude in NC (2 m) is greater than in LA (<0.5 m; *Wang, Lu & Sikora, 1993*), so we might expect snails from NC to climb higher if amplitude was engrained in their behavioral response. Thus, our observations of snail climbing was opposite the home marsh tidal amplitudes. The different heights between populations in the no predator treatments is unclear. Perhaps the smaller, diurnal tidal range which results in more prolonged periods of marsh flooding experienced in Gulf Coast marshes like LA (*Eleuterius & Eleuterius, 1979*) entrains local snail populations to remain higher when there is water present, since their ecological history suggests some predictability in predation risk. This would suggest some localized adaptation in the induced behavioral response (*Trussell & Smith, 2000*), and further support that different climbing heights is representative of a predator response, even if it is only a site-effect.

There are a number of mechanisms that might influence prey response to predators, including both physical and biotic, and unfortunately, these are difficult to elucidate without further experimentation and more sample sites along the geographic range of periwinkle snails. However, in our controlled setting, snails from LA consistently climbed higher than the NC snails. Biotic history and predator differences between the home marshes is a possible explanation for the differential behavioral responses. Although we did not measure crab abundance at the two collection sites, it is possible that there are differences between sites due to geographic locality (Fig. 4). Predation pressure varies biogeographically, with predation increasing as latitude decreases (*Bertness, Garrity & Levings, 1981*; *Peterson et al., 2001*), and numerous prey have
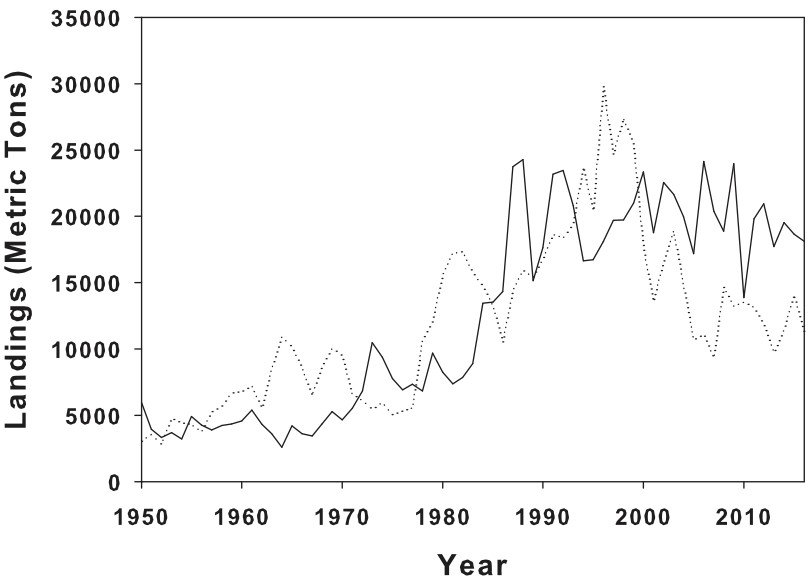

**Figure 4 Landings of blue crabs in Louisiana and North Carolina.** Blue crab landings from NOAA landings data for blue crabs in LA (solid line) and NC (dotted line) from 1950 to 2016.

responded by increasing defenses along this predation pressure gradient (*Bertness, Garrity & Levings, 1981*; *Freestone et al., 2011*; *Vermeij, 1982*). Further, predator identity and species composition, which can vary biogeographically, also lead to differential responses in prey species (*Large & Smee, 2013*). Multiple gastropods exhibited different avoidance behaviors across a broad temperate to tropical latitudinal gradient as a result of increased predator diversity (*Bertness, Garrity & Levings, 1981*). It is therefore possible that the observed differences in climbing height between the LA and NC populations of *L. irrorata* in our experiment might reflect differences in predation pressure experienced by the snails at their home marshes.

Unfortunately, it is difficult to make broad conclusions about geographic differences using only two study sites, and intraspecific trait variation could be due to a number of other factors that may vary independent of geographic location. For example, parasite load could reduce snail behavioral responses to predators (*Belgrad & Smith, 2014*), and it is unclear whether snails from either site had a higher parasite load which was beyond the scope of this experiment. Additionally, there could be other factors beyond predation pressure that could result in different climbing responses. While our periwinkles were offered food prior to the experiment, biotic history and tissue condition could play a role in response, as hungrier individuals may be more risky (*Gilliam & Fraser, 1987*), and other metrics of condition can alter activity (*Pardo & Johnson, 2004*). Although we controlled for hunger by feeding the snails while they were in captivity, the history prior to capture for this experiment could have played a role in site differences. Regrettably, we did not measure condition of the snails after experiments. Additionally, size and shell morphology (i.e., thickness, aperture size, spire length, etc.) might vary across locations for snails (*Sepulveda & Ibanez, 2012*; *Ramajo et al., 2013*; *Kosloski, Dietl &*

*Handley, 2016*), and could alter their escape responses to predators. Although we sought to use similarly-sized periwinkles from both locations, we did not measure morphometric variables. Finally, it is possible that even though blue crabs are common at both locations, we only used blue crabs collected locally in NC for our experiments, potentially leading to LA snails that were more alarmed by water- and airborne cues from a less familiar population of blue crabs, resulting in an exaggerated response. Regardless of the mechanism driving the differential responses, the snails collected in LA snails exhibited stronger responses to predators in our study system.

In addition, the results of this study demonstrate that periwinkles are likely responding to airborne predator cues, a novel observation for an intertidal, marine snail. When the crab was present but water was not allowed to exchange, there appeared to be an intermediate response in single population assemblages with LA snails, resulting in snails climbing ~5 cm higher than the no cue treatment snails. While this was not a statistically significant difference, it is the same size of the response in the NC snails with the waterborne cue. It is possible that a volatile compound given off by the crab can become aerosolized and perceived by the periwinkles. This has not been previously documented for marine organisms, however, chemosensory cues are common in terrestrial fauna, including gastropods (*Chase et al., 1978*; *Croll, 1983*), and a variety of aqueous compounds can be transported via the air, including HAB toxins (*Fleming, Backer & Baden, 2005*), as well as pyrazines, trimethylamine, and dimethyl sulfide (DMS; *Nevitt, 2000*). Terrestrial gastropods use airborne cues for homing, mating, and finding food (*Croll, 1983*; *Chelazzi, Le Voci & Parpagnoli, 1988*), as well as to avoid predators (*Bursztyka et al., 2013*; *Lefcort, Ben-Ami & Heller, 2006*). The response to airborne cues from predators has not been identified for other marine, intertidal snails, although, there is some indication that intertidal snails respond to airborne food (*Fratini, Vannini & Cannici, 2008*) and habitat cues (*Lazzeri, 2017*). Since marsh periwinkles spend much of their time out of the water, it is possible they could also be sensitive to airborne cues, and our experimental design was such that all of the snails could have been responding to airborne cues. It is beyond the scope of this experiment to determine which compound is becoming aerosolized and stimulating a response in periwinkles, but that such a chemical might exists warrants further attention.

We also note that snail response to cues only became significantly different when the two populations were mixed; that is regardless of origin, snails in mixed assemblages climbed at least twice as high as snails in single population assemblages in response to predator presence. This result is particularly interesting, because it suggests that some alteration in behavior might occur if distinct populations of the same species come into contact. While the probability of LA snails encountering NC snails in the field is low, rafted plant material can transport and disperse fauna great distances (*Thiel & Fraser, 2016*), tens to hundreds of kilometers (*Dame, 1982*; *Thiel & Gutow, 2005*; *Thiel & Fraser, 2016*). Further, climate change related distribution shifts (*Barry et al., 1995*; *Zacherl, Gaines & Lonhart, 2003*; *Mieszkowska et al., 2007*; *Poloczanska et al., 2013*) might lead populations of snails with different predator responses to interact with each other. Thus, it is increasingly likely that distinct populations with different predator responses can come into contact with each other. In the predator cue treatments, it is possible that

there was some avoidance between the populations, although the pattern of higher climbing was not observed in the mixed assemblage no cue control treatments. More likely, mixing the two populations together may have led to amplified alarm cues and signaling to other snails. Alarm cues are common, and while typically emitted from injured conspecifics (*Jacobsen & Stabell, 1999*), they could also come from disturbed, but undamaged, conspecifics (*Jacobsen & Stabell, 2004*). Alternatively, the periwinkles could have been responding to mucus trails of the other populations. Conspecific mucus trails have been shown to aid in navigation, homing, aggregation, and mating (*Ng et al., 2013*), and trails may also be a source of nutrition (*Davies & Beckwith, 1999*). Further, periwinkles may respond to both predator and alarm cues in mucus trails (*Duval, Calzetta & Rittschof, 1994*; *Ng et al., 2013*). The mechanism for the heightened response in mixed assemblages is unclear, and this experiment was unfortunately not designed to examine this. However, why this response might change in single population vs. mixed assemblages should be pursued in the future.

## CONCLUSIONS

This study demonstrates that geographic origin likely influences the behavioral response to a common predator for periwinkles. It is possible that the differential response to the common predator might be due to different abundance/distribution of blue crabs at the LA/NC home marshes, or just general trends of higher predator density/diversity with decreasing latitude. While further studies are required across more sites to ensure our observations are not just a site effect, these results are promising. In addition, this study is the first to demonstrate that these intertidal snails may also respond to airborne cues from predators. Although the chemical is unknown, that some volatile compound might become aerosolized and illicit a behavioral response in littorinid snails should be explored further. The broad geographic range and behavioral responses of *L. irrorata* make it a useful model organism to explore responses to waterborne, airborne, and even mucus-bound predator and alarm cues. Future work should investigate responses of snails from multiple locations along their biogeographic distribution, across multiple predator species which might also vary in abundance throughout the periwinkles' range, and identify compounds from predators and injured conspecifics that might become aerosolized.

## ACKNOWLEDGEMENTS

We would like to thank T. Bleier of the University of North Carolina Wilmington and other members of the Marine Biofluiddynamics and Ecology Lab (MARBEL) for help conducting lab experiments. We would also like to acknowledge Dr. Laura Treible of Georgia Southern for assistance in statistical analysis.

### Funding

This work was partially funded by UNCW and the Center for Support of Undergraduate Research and Fellowships (CSURF) at UNCW. There was no additional external or

internal funding for this research. The funders had no role in study design, data collection and analysis, decision to publish, or preparation of the manuscript.

### Grant Disclosures
The following grant information was disclosed by the authors:
UNCW and the Center for Support of Undergraduate Research and Fellowships (CSURF) at UNCW.

### Competing Interests
The authors declare that they have no competing interests.

### Author Contributions
- John M. Carroll analyzed the data, prepared figures and/or tables, authored or reviewed drafts of the paper, approved the final draft.
- Morgan B. Church conceived and designed the experiments, performed the experiments, authored or reviewed drafts of the paper, approved the final draft.
- Christopher M. Finelli conceived and designed the experiments, performed the experiments, analyzed the data, contributed reagents/materials/analysis tools, prepared figures and/or tables, authored or reviewed drafts of the paper, approved the final draft.

### Data Availability
The raw data are provided in a Supplemental File.

### Supplemental Information
Supplemental information for this article can be found online at http://dx.doi.org/10.7717/peerj.5744#supplemental-information.

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
