# Peer review of "Periwinkle climbing response to water- and airbone predator chemical cues may depend on home-marsh geography"

_PeerJ, doi:10.7717/peerj.5744_

## Round 0.1 · original submission · Major Revisions

The reviewers liked your paper but have raised issues that need addressing, including what you can validly conclude from your results. Please consider these issues in your revision.

Reviewer 1 ·

Basic reporting

The introduction could use some reorganizing and there are a few minor grammatical errors (See specific comments attached).

Experimental design

Many more methods details are necessary and I am concerned that the experimental design does not test for differences between airborne and waterborne cues but instead tests for differences in the concentration of airborne cues. Authors either need to revise manuscript to reflect this, provide methodological/observational details demonstrating that snails are actually responding to waterborne cues, or redo the experiment using a revised design (See specific comments attached).

Validity of the findings

Looking over the raw data the authors should have enough samples to test their hypotheses. The figures indicate that there are major differences between treatments and snail populations although revised statistical analyses are necessary. Conclusions are well stated and linked to original research question. However, some of the authors' conclusions on why snails exhibited different behaviors across geographical location need to be better identified as speculation given the experimental design and available data (See specific comments attached).

Annotated reviews are not available for download in order to protect the identity of reviewers who chose to remain anonymous.

Reviewer 2 ·

Basic reporting

The authors satisfactorily meet the standards of PeerJ

Experimental design

I have three concerns about the experimental design.

First, in both experiments, the LA and NC snails were exposed to NC crabs. Consequently, it is possible that the LA snails climbed higher than the NC snails because the LA snails were perhaps less familiar (and thus were more alarmed by) the water and air-borne cues of the NC blue crab. I understand that it would have been difficult to transport LA crabs to the laboratory in NC, therefore logistics likely contributed to the experimental design. But this is still a flaw in the design and interpretation of the experiment.

Second, the factorial nature of the second experiment is not appropriate. Based on my interpretation of the methods, the authors used tanks as the experimental unit. They randomly assigned tanks among the three cue treatments. However, they didn’t randomly assign tanks among the two levels of the snail factor (population/origin). Instead, they put both snail populations in the experimental unit. As a result, the two different populations represent two different response variables, not two different levels of an explanatory/independent factor.

Third, and I feel awful for bringing this up, but the authors collected snails form only one site in the high latitude and one site in the low latitude. Thus, it’s unclear if differences between the two populations is random or real. I think the authors should have collected snails from a few populations in the low and high latitude areas.

A few minor questions/comments:

Line 38: do you mean “evolved” instead of “involved”?

Line 60: melongena is the crown conch, not horse conch

What is the rationale for mixed population experiment? A rationale for this general topic (not snail specific) should be developed in the Introduction of the paper.

End of review

Validity of the findings

The findings have alternative explanations due to experimental design (see points 1-3 above). However, the authors conclusions are not unreasonable.

Additional comments

Manuscript 24555v1 “Periwinkle climbing response to predator chemical cues depends on home-marsh geography” submitted by Carroll et al. represents two laboratory experiments there were designed to test whether snails from a low latitude origin (LA) display stronger predator avoidance behavior compared to snails from a higher latitude origin (NC), presumably because predation pressure is higher at the lower latitude location.

The text is clearly written and the figures are simple as well as easy to interpret. In short, I enjoyed reading the manuscript and I appreciate the author’s efforts.

---

## Round 0.2 · Minor Revisions

The reviewers are much happier with this version of your MS, but have a few more relatively minor comments that need addressing in a final round of revision.

Reviewer 1 ·

Basic reporting

The authors have provided better context and background. The English is professional although there are a number of minor errors throughout the work. See annotated PDF.

Experimental design

Research question is well defined. The authors do a much better job highlighting and clarifying the limitations of their experimental design and discussing their results. Some minor method details still need to be added. See annotated PDF.

Validity of the findings

The statistical analyses needs to be slightly revised before publication. Speculation is better identified. See annotated PDF.

Annotated reviews are not available for download in order to protect the identity of reviewers who chose to remain anonymous.

Reviewer 2 ·

Basic reporting

The revised article is clear with sufficient background, figures and tables. The results are also relevant to the proposed research questions.

Experimental design

The revised draft addressed my criticism of their statistical analysis of the second experiment.

Validity of the findings

In the Discussion, I think it needs to be clearer that the significant differences among control and cue treatments did not manifest until the two snail populations were mixed.

Additional comments

The paper is about geographic origin of prey and multiple cues. I suggest modifying the title to reflect the multiple cue aspect.

At approximately line 70, the Introduction needs to highlight study from Brian Helmuth lab that demonstrated snail climbing is also due to wind speed, humidity, and temperature.

The idea of mixing populations and its relationship to the overall research connections needs a little more development. Intraspecific (trait) variation is a pretty popular research focus right now. I suggest highlighting this more.

End of review

---

## Round 0.3 · Minor Revisions

Thanks for submitting your revised MS. I'm happy to accept it after you make the following few very minor edits:

line 225: “climbing heights”

I can’t see where this has been addressed: “Lines 268-269: Presumably hunger level should not have been a factor since hunger level should have been standardized with your feeding regime since snails were all fed at the same time before the experiment start, yes? It should be mentioned in the methods that you controlled for hunger dependent behavior in this manner.
Yes, this is correct, and has been added to the methods.”

330: delete first “to” from following?: “might lead to populations of snails with different predator responses to interact with each other”

347: “influences”

---

## Round 0.4 · accepted · Accept

Thanks for making the last revisions.

#